# Minimization of Environmental Impact of Kraft Pulp Mill Effluents: Current Practices and Future Perspectives towards Sustainability

Gladys Vidal [1,*] , Yenifer González [1], Benjamín Piña [2], Mayra Jarpa [1] and Gloria Gómez [1]

[1] Environmental Engineering & Biotechnology Group (GIBA-UDEC), Environmental Science Faculty & EULA–CHILE Center, Universidad de Concepción, Concepción 4030000, Chile; yenigonzalez@udec.cl (Y.G.); mayjarpa@gmail.com (M.J.); gloriagomezosorio@gmail.com (G.G.)

[2] IDAEA-CSIC, Jordi Girona, 18, 08034 Barcelona, Spain; bpcbmc@cid.csic.es

* Correspondence: glvidal@udec.cl

**Abstract:** Kraft mill effluents are characterized by their content of suspended solids, organic matter and color due to the presence of lignin, lignin derivatives and tannins. Additionally, Kraft mill effluents contain adsorbable organic halogens and wood extractive compounds (resin acids, fatty acids, phytosterol) and show high conductivity due to the chemical compounds used in the digestion process of pulp. Currently, Kraft mills are operating under the concept of a linear economy and, therefore, their effluents are generating serious toxicity effects, detected in daphnia, fish and biosensors. These effluents are treated by activated sludge and moving bed biofilm systems that are unable to remove recalcitrant organic matter, color and biological activity (toxicity) from effluents. Moreover, under climate change, these environmental effects are being exacerbated and some mills have had to stop their operation when the flows of aquatic ecosystems are lower. The aim of this review is to discuss the treatment of Kraft pulp mill effluents and their impact regarding the current practices and future perspectives towards sustainability under climate change. Kraft pulp mill sustainability involves the closure of water circuits in order to recirculate water and reduce the environmental impact, as well as the implementation of advanced technology for these purposes.

**Keywords:** kraft mill effluents; toxicity; environmental impact; technologies; cycle closure; sustainability

## 1. Introduction

Latin America is leading the installed capacity of Kraft pulp mills, surpassing the installed capacity in North America. The continents that have a greater supply of cellulose fiber are Latin America and the Nordic countries, while the largest demand for cellulose fiber comes from China. The developing countries that install this type of Kraft pulp mill generate processes with the best available technology (BAT) in the market, that is, with continuous pulp digestion systems and elemental chlorine-free (ECF) bleaching pulp. Additionally, developing countries have conventional treatment systems installed and then discharge their effluents to surface ecosystems (i.e., rivers or sea). Developed countries face growing conservationist pressures that limit or make it more difficult to obtain raw materials for industry. Furthermore, in these countries, there are still very competitive locations for the development of plantations; the requirements for forest management certifications and sustainability codes are universal for all countries that produce and export pulp. On the other hand, Europe is no longer the most important regional market, being displaced by China, and pulp based on eucalyptus continues to lead the market growth, adding 3.1 million tons of annual growth.

In this article, for a better understanding of the difficulty that the Kraft pulp mills have in order to implement cycle closure, the origin of the effluent, its generation under BAT, and

adequate procedures that all industries that export pulp have implemented, as required by international markets, are explained. Due to the raw material and the chemical conditions of the process, it generates a series of environmental impacts that have been evaluated by means of different bioindicators at different trophic levels. The conventional treatment technologies that are currently installed in all Kraft pulp mills are of the conventional, biological type and all discharge their effluents to surface water bodies. However, the water crisis that is affecting many countries that produce Kraft pulp requires sustainability option to increase the level and type of effluent treatment using advanced technology, so as to be able to close certain parts of the production process and, with them, reduce the use of fresh water.

The aim of this review is to discuss the treatment of Kraft pulp mill effluents and their impact regarding the current practices and future perspectives towards sustainability under climate change.

## 2. Kraft Mill Process and Effluents

Wood is known to be the most abundant and renewable source of lignocellulosic material in the world. It has three main components: cellulose (40–45%), hemicelluloses (20–30%), and lignin (20–30%) [1]. Additionally, other organic compounds (2–5%) can be extracted from digestion processes, including terpenes, polar phenols, fatty acids, resin acid, and sterols. These are known as resins, extractive compounds, or wood extractives [2].

Long fiber wood, such as pine (i.e., *Pinus radiata*), presents a greater quantity of wood extractives (0.5–7.0%), while short fiber wood (eucalyptus, i.e., *Eucaliptus globulus*, *Eucaliptus nintens*) present a lower quantity (0.2–3.5%) [3]. Long fiber wood is rich in resin acid [4], while sterols are abundant in short fiber wood [5]. However, the effluents of both long and short fiber woods contain sterols, resin acid, long chain fatty acids, and other compounds. Processes that use short fiber or mixed fiber wood produce a greater extractive compound load in their effluents [4,5]. On the other hand, studies have demonstrated that compounds with log Kow (octanol–water partition coefficient) values higher than 4 have accumulation properties and, because of this, probably biological activity. Specifically, Kraft pulp mill effluents generate micropollutants with high Kow, as is the case of stigmasterol (10.2), β-sitosterol (9.6), abietic acid (4.6–7.5), and dehydroabietic acid (5.7–7.2), among others [6].

On the other hand, the production process of Kraft pulp mills generates different effluents whose physicochemical characteristics depend on the raw material, technology, and processes used. The processes that generate most of the pollutant effluent are digestion and bleaching. Due to this, the best available technologies are installed to increase properties in the pulp and to reduce the absorbable organic halogen (AOX) concentration in the effluent, due to the bleaching process.

Figure 1 summarizes the main compounds that are produced during the main stages of the process of obtaining Kraft cellulose [7].

The main strategies for minimizing effluent generation and its effect are: (a) increased delignification efficiency in earlier stages, which represents a longer bleaching stage, and (b) improved technical conditions for pulp bleaching. The following procedures have been proposed: (1) substituting chloride for another oxidating agent, such as oxygen or chloride dioxide [8,9], (2) water recycling, (3) modifying the washing system [10], and (4) performing the bleaching stage with pulp of greater consistency in order to save water.

All the technological improvements mentioned above will help to improve biodegradation in the effluents, reducing their environmental impact due the biological activity of micropollutants, among others. However, the micropollutants in the generated effluents need to be studied. For example, bleaching technology that does not use chlorates as oxidants (total chlorine-free (TCF)) generates effluents with a smaller contamination load than the effluents produced by elemental chlorine-free (ECF) bleaching processes [11] due to the organic compounds and the highly oxidized compounds present in these effluents [12]. However, the TCF process effluents require chelants such as ethylenediaminetetraacetic

acid (EDTA) and diethylenetriaminepentaacetic acid (DTPA) which show low biodegradation in treatment plants and, consequently, high incidence in the environment [13,14].

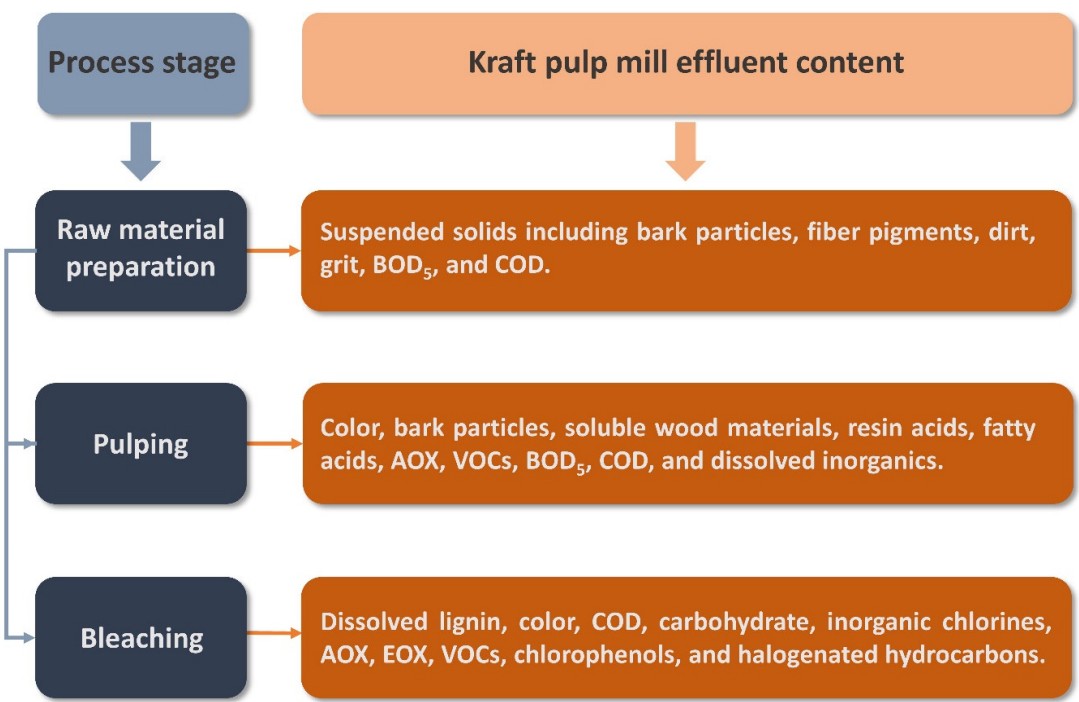

**Figure 1.** Major pollutants released from Kraft pulp mill effluent. Source: adapted from Kamali and Khodaparast [7]. $BOD_5$: biological oxygen demand; COD: chemical oxygen demand; AOX: adsorbable organic halogens; VOC: volatile organic compounds; EOX: extractable organic halogens.

## 3. Recalcitrant Organic Compounds in Kraft Pulp Mill Effluents

Kraft pulp mill effluents contain a variety of recalcitrant compounds, such as lignosulfonic acid, chlorinated resin acids, chlorinated phenols, dioxins, and chlorinated hydrocarbons. Figure 2 shows a type of classification of organic matter and the possible compound removal by physicochemical, chemical, and biological technology.

Although in most cases the toxicity is low, pulp bleaching effluents are characterized by a high concentration of chemical oxygen demand (COD) (1000 to 7000 mg/L), a low biodegradability ratio (biological oxygen demand, $BOD_5$/COD) of 0.02 to 0.07, and a moderate concentration of suspended solids (500 to 2000 mg/L). Compounds, especially those containing chlorine (measured by the parameter AOX) are recalcitrant because they contain chemical structures that are rare in nature, such as the carbon–chlorine bond. It has been widely reported that high molecular weight organic matter (HMW >1 kDa) in bleaching effluents is more recalcitrant to biological treatment than low molecular weight organic matter (LMW <1 kDa) [15,16]. Dissolved lignin and its degradation products, hemicelluloses, resin acids, fatty acids, diterpenic alcohols, juveniles, tannins, and phenols are responsible for the dark color and toxicity of the effluent [17].

Lignin and its derivatives are recalcitrant and highly toxic compounds responsible for the high $BOD_5$ and COD values of the effluents, as well as the dark brown color of the pulp effluents formed during pulping. Lignin is one of the most difficult substances to break down [18]. For example, lignosulfonates have been found to inhibit various biological systems, such as enzyme, toxin and antibiotic functions, and the chlorinated derivatives of lignin are poorly degraded by conventional wastewater treatment processes. During the biological and chemical degradation of chlorinated lignin, small (MW <1 kDa) harmful compounds can be formed, such as chloroanisoles and chloroveratrols, which, when accumulated in fish, can cause a bad taste. In addition, the concern is that chlorinated lignins release toxic or bioaccumulative compounds or are transformed into biologically

active compounds [19]. Table 1 shows the physical–chemical characterization of some wood extractives present in effluents from the Kraft pulp industry.

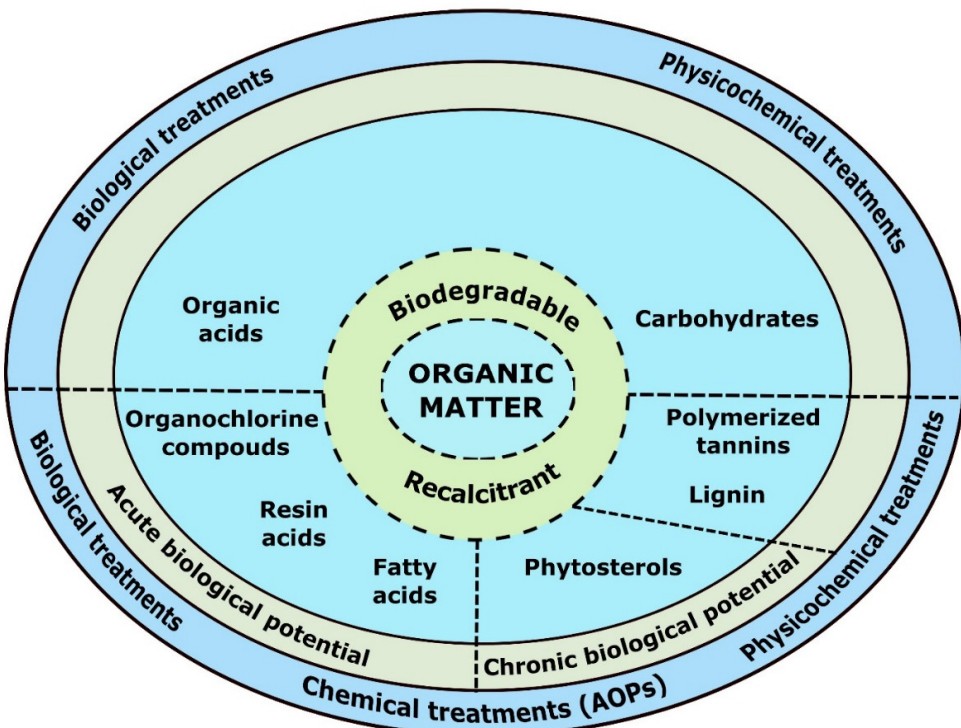

**Figure 2.** Schematic representation of the biodegradability and recalcitrance of the organic matter contained in Kraft mill effluent and its treatment feasibility using different technologies.

**Table 1.** Physicochemical properties of some wood extractives with biological activity contained in the Kraft pulp mill effluents.

| Compound | Structure | Molecular Weight (g/mol) | Solubility (mg/L) 20 °C | Log Kow | Reference |
|---|---|---|---|---|---|
| Abietic acid | | 302.46 | 4.75 mg/L | 4.6–7.15 | [20] |
| Dehydroabietic acid | | 300.44 | 5.1 mg/L | 5.7–7.25 | [20–22] |
| Campesterol | | 400.69 | <0.0001 | n.r. | [23,24] |
| Stigmasterol | | 412.70 | <0.0001 | 10.20 | [23–25] |
| β-sitosterol | | 414.72 | <0.0001 | 9.65 | [23,24,26] |
| Genistein | | 270.241 | n.r. | 5.9 | [27,28] |

n.r.: not registered.

## 4. Evaluation of Biological Activity Effects of the Kraft Mill Effluents in the Environment

The compounds present in Kraft mill effluent discharges in the environment that can potentially interrupt the normal functioning of endocrine systems of the biota in an ecosystem have motivated numerous investigations because the effects on human beings are not yet evident [29]. The traditional method of evaluating these compounds in industrial effluents is by using chemical methods, such as spectrometry (UV–visible, IR) and chromatography (thin layer chromatography (TLC), high-performance liquid chromatography (HPLC)), and gas chromatography–mass spectrometry (CG-MS), which are analytical techniques of great detection power for chemical compound quantification and identification, but their operation is very complex and expensive [30]. On the other hand, potential biological effects of Kraft mill effluents have been studied in experimental exposure experiments in fish [29,31] or *Daphnia magna* [6,32,33]. Specifically in this case, the research group has detected estrogenic activity of Kraft mill treated effluent by a recombinant *Saccharomyces cerevisiae* yeast. The results indicate that the estrogenic activity values were relatively low, between 1.475 and 0.383 ng/L of estrogenic equivalent of 17 a-ethynylestradiol (EE2 eq.), where the highest value corresponds to the *Eucaliptus globulus* effluent and the lowest value to the *Pinus radiata* effluent [30,34]. However, previous works have found toxic effects in treated Kraft mill effluents due to dioxin-like compounds; in other words, ligands of the aryl hydrocarbon receptor (AhR) constitute a significant fraction of the biological activity of Kraft mill effluents [35]. Ectopic activation of AhR constitutes the initial step of the metabolic chain, leading to toxic effects of a variety of different harmful pollutants, such as 2,3,7,8-tetrachlorodibenzo-(p)-dioxin (TCDD) and benzo[a]pyrene (BaP). Immune dysfunction, endocrine disruption, reproductive toxicity, developmental defects, and cancer in vertebrates are some of these effects [36]. Therefore, assays to detect AhR activation and subsequent signal transduction are becoming an extremely useful approach to monitor pollution loads in environmental samples. The advantage in using this biosensor is that yeast is easy to grow and detects chemical compounds rapidly and at a low cost. The application of this test in the field of wastewater treatment is in full expansion. To date, it has been used widely in the evaluation of wastewater treatment plants [3,30,37].

Depending on the effects that these tests need to detect, organisms, cells, and/or tissues can be used. The simplest and best known are the acute toxicity tests that have been developed and standardized by US agencies such as the United States Environmental Protection Agency (USEPA). These toxicity tests can be used to determine a contaminant's capacity (whether a pure substance or effluent) to produce toxic effects in live organisms when they are exposed for a certain time at certain concentrations. The most used organisms in aquatic biotests are daphnia, mainly due to their wide distribution (cosmopolitan), their ecological importance, their sensitivity to disrupted environments, and their short life cycle. As a result, they are considered indicator species for adverse environmental conditions [38]. The adequate survival, growth, and reproduction of daphnia are crucial for success in the environment. Moreover, studies on this species could give information on the effluent quality in terms of the ecosystem trophic chain [39]. Some studies have focused on evaluating the effects of endocrine-disrupting chemicals on daphnia. Thus, Xavier et al. [32] found that Kraft mill effluents induce sexual maturation in female daphnia. The development of secondary sex characteristics has been demonstrated to be altered by endocrine-disrupting chemicals. On the other hand, Olmstead and Le Blanc [40] observed that diethylstilbestrol stimulates the development of the secondary abdominal process in female daphnia. Moreover, studies showed that the effluent factors capable of modifying the body proportions of daphnia have the same effect in pine and eucalyptus. Additionally, they found that micropollutants like β-sitosterol and stigmasterol contribute to the allometric growth rate (determined as % of the growth rate of the body length and body width). The phytosterols per se are responsible for 12.9 and 8.1% of the deviation from the natural shape, while the Kraft mill effluents account for 25.6 to 27.8% of shape deviation. They concluded that the estrogenic activity of Kraft mill effluents is dependent

on the species processed for wood pulp, but the effluent treatment and the operation strategy were not evaluated in this work.

A second biological response widely found in fish exposed to bleached Kraft mill effluents is increased detoxification activity in the liver and other tissues [41–43]. The most widely used marker for this biological activity is the activation of cytochrome P450 1A (cyp1a), monitored through the associated enzymatic activity of ethoxyresorufin O-deethylase (EROD) [44,45] or analyzing changes in the transcription of the cyp1a gene using real-time polymerase chain reaction techniques [46–48]. This activity, also called "activity similar to dioxins", is not considered an adverse outcome as such, although it constitutes the initial step in the metabolic chain leading to the toxic effects of a variety of harmful contaminants, such as 2,3,7,8-tetrachlorodibenzo-(p)-dioxin (TCDD), coplanar PCBs, and benzopyrenes, among others [36]. It is important to emphasize that this response is mediated by the interaction of contaminants with a specific nuclear receptor, the aryl hydrocarbon receptor (AhR), also known as the "dioxin receptor", with which mice lacking the corresponding gene become resistant to the tumorigenic effects of dioxins or benzopyrenes [49,50].

Table 2 summarizes some effects on the biological activity of some species exposed to effluents or compounds from Kraft pulp mills. Additionally, the means of detection are indicated.

**Table 2.** Effects on the biological activity of some species exposed to effluents or compounds from Kraft pulp mills.

| Exposed Species | Effluent/Compound | Toxicity Effect | Detection | Reference |
|---|---|---|---|---|
| *Gambusia affinis* | BKME | Gonopod formation (female) | Body measurements | [51] |
| *Oryzias latipes* | Genistein (1000 µg/L) | Intersex at 12%. Large ovarian lumen | Histological analysis | [27] |
| *Trichomycterus areolatus* and *Percilia irwini* | Bío-Bío river: downstream and upstream pulp mill effluent discharges | Higher level of VTG and EROD in fish exposed to downstream pulp mill effluent discharges. Additionally, gonad alterations and intersex juvenile fish | Western blot, Northern blot. VTG, EROD, LSI, GSI, GC-MS, ELISA | [29] |
| *Coregonus lavaretus* | BKME-ECF (5 mg/fish) | Induction of VTG mRNA | Northern blot | [52] |
| *Daphnia magna* | BKME (6.25, 12.5, 25, 50, 100%) | Abdominal growth | Toxicity tests, CG-MS, microscopy | [32] |
| *Carassius carassius* | β-sitosterol (200 mg/g) | Reduction in the size of the gonads | VTG, GSI, HSI and histological analysis | [53] |
| *Danio rerio* | DHAA (50 µg/L) (F0 y F1) | VTG increase (F1 males); low VTG levels (F0 males) | VTG, ELISA, and histological analysis | [54] |
| Sprague Dawley rats | Genistein (12.5, 25, 50, 100 mg/kg) | VTG increase (F1 males); low VTG levels (F0 males) Females: irregularities in the heat cycle, histopathological changes in the ovaries and uterus, loss of fertility (100 mg/kg) | Histological analysis | [55] |
| *Selenastrum capricornutum, Lemna aequinoctialis* | BKME-ECF | 7-day growth | 7-day growth | [56] |

VTG: vitellogenin; EROD: ethoxyresorufin-o-deethylase; ELISA: enzyme-linked immunosorbent assay; BKME: bleaching kraft mill effluent; ECF: elementary chlorine free; GC-MS: gas chromatography–mass spectrometry; GSI: geological strength index; HSI: heat stress index; DHAA: dehydroabietic acid. Adapted from Monsálvez et al. [28].

Table 2 shows the toxicity of the effluents from the Kraft pulp industry. Many studies have found toxicity in the effluents that are discharged to surface bodies. This means that there are bioactive compounds that are not eliminated by the conventional secondary treatments that are currently installed in the production processes. Table 1 shows examples of extractive compounds of difficult biodegradability that may be present in effluent discharges (i.e., campesterol, stigmasterol, β-sitosterol, genistein). All these compounds share basic phenolic structures that are difficult to biodegrade, joined by double bonds.

In addition, biological potential studies on these effluents have used the recombinant *Saccharomyces cerevisiae* biosensor (yeast estrogen assay, YES). Therefore, if this strain is genetically modified with the human estrogen receptor (hER), a recombinant yeast assay (ER-RYA) bioindicator can be obtained and if YES is genetically modified with the human aryl hydrocarbon receptor (AhR), it is possible to detect compounds with structures similar to dioxins and furans (YCM-RYA) [30]. Chamorro et al. [57] studied three Kraft mill effluents with different raw materials (P. radiata, E. globulus, and their mixture: mixed),

detecting estrogenic activity expressed as 17-α-ethinylestradiol equivalent (EE2 eq.) of 0.383 ng EE2 Eq/L, 1.475 ng EE2 Eq/L, and 0.849 ng EE2 Eq/L, for effluents of P. radiata, E. globulus, and mixed, respectively, Fernández et al. [58] obtained values of 42–83 ng EE2 Eq/L. In the case of ER-RYA (estrogenic activity) and YCM-RYA ("dioxin-like" activity) studies, carried out by Monsálvez et al. [28], estrogenic activity values expressed as 17-β-estradiol (E2 eq.) were from 0.19–0.68 ng E2 Eq/L and 0.28–0.67 ng E2 Eq/L for effluents of P. radiata and E. globulus, respectively, and for dioxin-like activity, values of 21.35 ng E2 Eq/L and 753.80 ng E2/L were found for effluents of P. radiata and E. globulus, respectively, while Chamorro et al. [30], in sediment samples near Kraft cellulose discharges, detected low levels of responses for YES and RYA.

In addition to this, the effects of aquatic toxicity generated by Kraft mill effluent discharges may produce a direct or indirect influence on the food chain related to productive activities, such as agriculture and fishing. Moreover, toxic compound discharges are unsuitable for the growth and development of microbes, plankton, and small fish. This further affects the growth of larger fish. Furthermore, the accumulation of toxic materials, such as polyhydroxybutyrates and persistent organic pollutants, makes them toxic to secondary and tertiary consumers, causing health problems.

## 5. Kraft Pulp Mill Effluents Treated by Conventional Technologies

The biological aerobic treatments with suspended biomass most commonly used in Kraft mills are: aerated lagoons, activated sludge (AS) [59], and moving bed biofilm reactors (MBBRs) [60,61].

Aerated lagoons are easy to operate but require a long hydraulic retention time (HRT) and elevated land extensions. Additionally, aerated lagoons have problems in separating the generated excess of solids. Furthermore, the operating conditions strongly influence the degradation of aromatic compounds [62]. Xavier et al. [59] demonstrated that in anoxic areas, intermediate compounds of resin acid biodegradation (e.g., retene) can produce disruption activity. However, in optimal operating conditions, the biodegradation of these compounds can be greater than 90% [22]. Aerobic treatment of Kraft mill effluents by aerated lagoon systems reduces COD from 35–50%, $BOD_5$ up to 90%, and suspended solids 80% [22,59]. However, organic compounds with high molecular weight and recalcitrant compounds are partially transformed by aerobic bacteria (biotransformation) without reaching complete mineralization of organic matter to $CO_2$ and $H_2O$ [62]. However, at an organic load rate (OLR) greater than 2 g COD/L·d (food/microorganisms (F/M) = 0.56 g COD/g VSS·d), phytosterol removal was 66.5%, although there was no removal of either acute toxicity (median lethal concentration, LC50, 48 h = 88.22%) or chronic reproduction and growth toxicity (lowest observed effect concentration, LOEC = 20%), whereas genotoxicity increased 16% [59]. Optimal operation of aerobic treatment systems depends on the adequate control of operational parameters, such as: organic load rate, pH, temperature, and aeration [59–61,63].

On the other hand, activated sludge systems used to aerobically treat Kraft mill effluents at the industrial scale remove 50% of COD in effluents from softwood processing and more than 65% of COD in effluents from hardwood processing industries [63]. However, some treatments do not remove the effluent color efficiently, and in other cases they even increase it. Successful removal of extractive compounds (up to 97%) can be verified in activated sludge, but it is not totally clear if the removal of specific compounds, such as resin acid (43–94%) and phytosterols (41–99%), is due to biotransformation or adsorption in the sludge [64]. However, the treatment of eucalyptus effluent removes almost 64% of the total sterol content in the primary treatment, while 36% is passed to the activated sludge reactor. Of these contents, between 41 and 67% were biodegraded or biotransformed in the biological system, and between 31 and 57% were removed by adsorption in the sludge and then subsequently thickened and disposed of [64]. Mahmood-Khan and Hall [26] observed that β-sitosterol and β-sitostanol are the most removed phytosterols when Kraft cellulose effluents are biologically treated (60–80%).

In the effluents coming from processes using eucalyptus as raw material, a higher sterol concentration is found, in which β-sitosterol presents the highest proportion (up to 34 g ptp) [34]. The secondary treatments of the nine plants in this study presented an elevated efficiency in resin acid and sterol degradation. However, the final system's removal efficiency can vary between 53 and 99%. This study also demonstrated that the resin acid and saturated and unsaturated acid concentrations found in the effluent depend on the type of wood used as raw material. From the point of view of the treatment of these compounds, the volatile unsaturated fatty acids present a higher degradation percentage than the saturated fatty acids. Xavier et al. [32], in a comparative study, show that an activated sludge system presents greater elimination efficiency for compounds with estrogenic activity in an aerated lagoon. Similarly, the removal of phytosterols in conventional activated sludge systems operating at an OLR of 9.0 gCOD/L·d and HRT of 3 h was 70.3%. Still, even though the acute and chronic toxicity were completely removed, the genotoxic effect increased 6%. At low F/M ratios, the biomass sedimentation in activated sludge was affected [60].

On the other hand, MBBR systems can operate at an HRT of less than 2 h and their operation can be extended to nitrogen and phosphorus removal. As a result, an MBBR system can use 1/5–1/10 of the space occupied by a conventional sludge system [60]. The biofilm used in this type of system plays an essential role in the system efficiency. The transfer of oxygen and/or nutrients can be a limiting factor for biofilm growth and for system robustness. Numerous studies have been performed in which the best operational conditions are evaluated for systems using this type of technology, depending on the substrate used, and there have been studies that evaluate the support together with the operating conditions [63]. Specifically, AnoxKaldnes studied the behavior of a biofilm that grows on a support for systems that operate in the first stage of the biological treatment system (higher $BOD_5$ load), and hybrid systems or systems that use it as a polishing stage [60,61]. Table 3 shows the removal performance of conventional technologies for organic matter and active compounds contained in the Kraft pulp mill effluents.

**Table 3.** Performance of the organic matter and active compounds contained in the Kraft pulp mill effluents treated by conventional technologies.

| Technology | HRT (h) | OLR (kgBOD$_5$/m$^3$·d) | BOD$_5$ (%) | COD (%) | Resin Acid (%) | Phytoesterols (%) |
|---|---|---|---|---|---|---|
| Aerated lagoon | 480–48 | 0.01–0.2 | 85–96 | 42–55 | 50-97 | 61–78 |
| Activated sludge | 48–4.5 | 0.4–1.4 | 85–99 | 42–93 | 80-99 | 50–98 |
| MBBR | 1.7–3 | 0.3–10 | 75–99 | 60–90 | 85-99 | 98–99 |

MBBR: moving bed biofilm reactor; HRT: hydraulic retention time; OLR: organic load rate; BOD$_5$: biological oxygen demand; COD: chemical oxygen demand; AOX: adsorbable organic halogens [3,22,57,59–61].

## 6. Advanced Treatments Used in Kraft Pulp Mill Effluent Treatments

Advanced oxidation processes (AOPs) and membrane technologies have emerged as an alternative to conventional technologies for the oxidation of recalcitrant compounds. These processes are based on the generation of hydroxyl radicals, which are strong oxidants for the complete mineralization of the target compounds.

Table 4 summarizes studies carried out both in physical and physical–chemical treatment systems, such as chemical precipitation and chemical treatments, such as AOPs, all of them compared regarding COD, TOC, and AOX removal in Kraft mill effluents. Thus, for chemical precipitation, the removal of COD is 63–77%, while for TOC it is 30–70%, with very few studies regarding active compounds, such as phytosterols, with a removal greater than 90% for β-sitosterol and stigmastanol [65]. In the case of AOPs, the COD and TOC removal range from 20–94% and 8–96%, respectively [66]. In addition to this, the removal efficiencies of resinic acids, linoleic acid, and β-sitosterol were 36–93%, 84%, and 87%, respectively, by the technologies of $UV/H_2O_2/Fe^{+2}$ and $O_3$ [65].

**Table 4.** Efficiencies in different physicochemical and chemical treatment technologies used with Kraft pulp mill effluents.

| Technology | COD (%) | TOC (%) | Color (%) | Phenolic Compounds (%) | Active Compounds (%) | Reference |
|---|---|---|---|---|---|---|
| Physicochemical technology | | | | | | |
| Chemical precipitation | 63–77 | 30–70 | 96 | n.r. | >90 [a,b] | [65,67,68] |
| Chemical technology | | | | | | |
| $UV/H_2O_2$ | 74 | 8–45 | 41 | 24–91 | n.r. | [69,70] |
| $H_2O_2/Fe^{+2}$ | >60 | 20–90 | 85 | 32–100 | n.r. | [69,71] |
| $UV/H_2O_2/Fe^{+2}$ | n.r. | 60–96 | 82 | n.r. | 93 [a], 84 [c], 97 [d] | [65,69,71] |
| $O_3$ | 29–76 | 19–51 | 81–97 | 85–100 | 36–90 [c] | [33,72,73] |
| $O_3/H_2O_2$ | 31 | n.r. | 81 | 58–93 | n.r. | [69,74] |
| $O_3/UV$ | 20 | n.r. | 30 | 81–93 | n.r. | [70,75] |
| $UV/Zn$ | 69–94 | 80 | n.r. | n.r. | n.r. | [76] |
| $UV/TiO_2$ | 75–80 | n.r. | n.r. | 42–78 | n.r. | [70,77] |
| $O_3/UV/ZnO$; $O_2/UV/Zn$ | 50 | n.r. | 40 | n.r. | n.r. | [75] |
| Physical technology | | | | | | |
| Reverse osmosis | 89 | n.r. | 100 | n.r. | n.r. | [78] |
| Ultrafiltration | n.r. | n.r. | 92 | n.r. | 72 [e] | [79] |
| Nanofiltration | n.r. | n.r. | 72 | n.r. | 82 [e], 100 [f] | [79,80] |

n.r: not registered; a: β-sitosterol; b: stigmastanol; c: resinic acids; d: linoleic acid; e: AOX: adsorbable organic halogens; f: endocrine-disrupting activity. COD: chemical organic demand; TOC: total organic carbon.

On the other hand, ozonation as a unit treatment has proven to be a strong disinfectant and capable of eliminating color and oxidizing recalcitrant compounds without altering the toxicity of the treated effluent, due to the total mineralization of the compounds [65,81]. Mainardis et al. [82] showed that ozone treatment could effectively replace tertiary physicochemical treatment in terms of COD and TSS elimination, which would mean an economic saving of EUR 300,000/year and the investment could be recovered in approximately 7 years.

Furthermore, within the AOPs are photoelectrocatalysis (PEC) processes, which arise from the combination of photocatalysis (PC) and electrochemical (CE) processes. This technology has been evaluated little for Kraft mill effluents, while it is sustainable because it can compensate for the high electrical energy consumption of the EC processes and the input of external current in the PEC systems. In these systems, $TiO_2$ is one of the most widely used and most studied photoanode materials due to its non-toxicity, low cost, and strong oxidizing capacity [17,83]. Rajput et al. [84] found that $TiO_2$ electrodes together with Au improve the photoelectrocatalytic activity of $TiO_2$ electrodes, producing a 63.5% reduction in COD and 44.4% in TOC.

Systems based on membrane filtration have been shown to have a high removal of color, COD, AOX, salts, heavy metals, and total dissolved solids. These processes can range from microfiltration (MF), ultrafiltration (UF), and nanofiltration (NF) to reverse osmosis (RO) [19]. Through reverse osmosis, it was possible to obtain a maximum removal of 88% of $BOD_5$ and 89% of COD [78], with the typical efficiency of membrane technologies being 50 to 90%. On the other hand, regarding organochlorinated compounds, the highest removal of AOX and color achieved by ultrafiltration was 72 and 92%, respectively. Meanwhile, the total removal of color and more than 90% removal of AOX are achieved by nanofiltration [79]. Moreover, Salvaterra et al. [80] show that nanofiltration is able to prevent endocrine-disrupting activity. The membrane used as a tertiary treatment could contribute to removing organic compounds contained in the bleaching effluents with the possibility to reuse the effluent within the process, thus reducing discharges with active micropollutants to aquatic ecosystems.

## 7. Towards a Circular Economy and Sustainability in Kraft Pulp Mills: Perspectives

The pulp and paper industry has received much criticism from all over the world, particularly from environmentalist groups. Lumber harvesting for the paper industry has been linked to increased deforestation in the world's forests. On the other hand, when a monoculture of pine and/or eucalyptus plantations is introduced, water stress occurs

in the disrupted hydrographic basins, generating great social conflicts due to pressure on the drinking water sources. This type of industry generates two different types of pressure on the water of an ecosystem. On the one hand, due to the consumption of large amounts of water by metabolism due to the requirements of the plant, the water that is taken underground is evapotranspired by the plants and then the amount of surface water is reduced. On the other hand, the small amount of surface water available is not enough to then dilute the components of the effluents that are discharged from Kraft pulp mill processing.

Under a climate change scenario, water is a scarce resource and an element of social conflict due to the various uses that an ecosystem must provide. Currently, the Kraft pulp industry is working in an open circuit. That is, it takes water from ecosystems, uses it in the technological process and then discharges the treated effluent to surface ecosystems. However, the sustainability of the industry in terms of water resources can be a problem in the short and medium term for two reasons: (a) insufficient water in the ecosystem to feed the production process and (b) insufficient flow for aquatic ecosystems to promote dilution of the effluents to be discharged.

For this reason, it is very important to evaluate quaternary treatment systems that allow a final effluent quality that mean it can be reused while maintaining the stable operation of the process. The recirculation of water, under a closed cycle, in the Kraft pulp industry must include not only the removal of organic compounds by conventional technology, but also ions dissolved (non-process elements) in a concentration such that they do not generate problems, such as incrustations, corrosion, and quality problems in the final product due to their accumulation. Therefore, it is desirable to explore the coupling of various biological, physical, and chemical technologies, thinking about the recovery of water, energy, and valuable or undesirable compounds for these production processes. Figure 3 shows the projection of the operation of a Kraft pulp mill operating in an open circuit to a closed cycle. Currently, this considers the scientific evidence of current impacts on ecosystems. On the other hand, it shows the possibility that these processes can close the cycle, changing to closed-cycle systems due to the introduction of quaternary technologies of a physicochemical type.

Currently, there are no (or not publicly known) bleached Kraft pulp plants that reuse all of their effluents [85]. However, reverse osmosis (RO) is the most versatile desalination method for the treatment of water of any salinity, from brackish water to high-salinity water, linking non-process elements from Kraft pulp mill effluents [86]. The main weaknesses are the fouling of the membranes due to the action of organic substances, which can be solved with an adequate pre-treatment, and the formation of crystals that are embedded in the membranes, which affect their performance and limit the recovery achieved [87]. None of these options have been applied on an industrial scale to treat bleached Kraft pulp mill effluents.

On the other hand, electrodialysis (ED) is an electrochemical separation process which employs electrically charged ion exchange membranes with an electrical potential difference as a driving force. In electrodialysis, ions in solution migrate through ion-selective membranes in an electric field [86,87]. The membranes are impervious to water, so, unlike RO, ED is based on promoting selective ion transport rather than selective transport of water. However, the energy consumption is mainly due to the electric field, while in RO, energy is used to pump the water through the membranes. A variation of ED is EDR, in which the polarity of the electrodes is periodically reversed (the hydraulic polarity is also reversed), generating an exchange of the concentrated cells for the diluted cells and vice versa. This technology promotes the separation of particles deposited on the membranes, automatic cleaning of the system, and greater resistance to fouling by organic contaminants or salt scale, compared to RO. This makes it a better option in the face of adverse conditions, such as Kraft mill effluents. Regarding the energy consumption, it is similar to RO, although it depends on the concentration of salts in the solution.

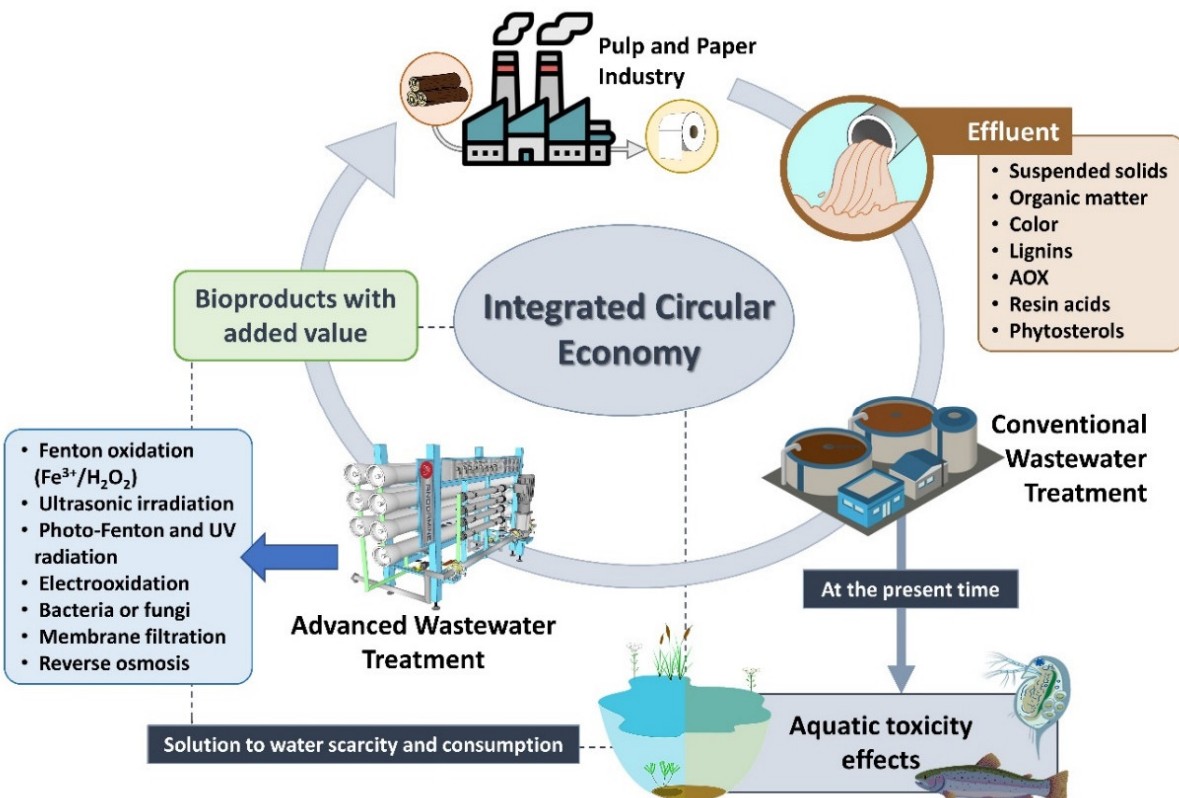

**Figure 3.** Integrated circular economy in the Kraft pulp mill.

The circular economy will promote the biorefinery of lignocellulosic compounds. In developing countries, where the raw material has low value, the biorefinery of lignocellulosic compounds has not had a decided impact on basic and applied research regarding biorefinery of lignocellulosic compounds. The market has only given space to the main bleached Kraft pulp product. However, under climate change and the new production scenarios of water scarcity, the research of the current biorefinery concepts should be elaborated and optimized for the integrated utilization of all products in high-value applications rather than focusing on bioethanol, biofuels, or sugars only, as is still common to date [88]. In particular, the engineering of lignin is not studied. Nevertheless, considering the crucial role of lignin and other biomass components, the development of the different biotechnologies of biomass and lignin, in particular, are very promising research fields, in light of the multiple future applications, such as fibers, nanofibers, nanoparticles, and products, among many others not yet known. In addition to that, parallel development of the biotechnological potential of plant-associated microorganisms will be carried out [89].

These include technical issues associated with integrating operation units with each other, integrating production of individual products into a multi-product biorefinery, and integrating biorefineries into the broader resource, economic, and environmental systems in which they function. We anticipate that coproduction of multiple products, for example, production of fuels, chemicals, power, and/or feed, is likely to be essential for economic viability. Lifecycle analysis is necessary to verify the sustainability and environmental quality benefits of a particular biocommodity product or process. We see biocommodity engineering as a legitimate focus for graduate study, which is responsive to an established personnel demand in an industry that is expected to grow in the future. Graduate study in biocommodity engineering is supported by a distinctive blend of intellectual elements, including biotechnology, process engineering, and resource and environmental systems [90].

## 8. Conclusions

The consequences of climate change and water scarcity are forcing Kraft pulp companies to investigate alternatives to managing water, moving from an open cycle to a closed cycle. Due to the chemical characteristics of these effluents and as the currently installed technology generates final effluents with a fraction of recalcitrant organic matter, it is important to investigate technology that is coupled to the currently installed biological technologies. The separation of some effluents at the source, as well as the intensification of the treatment of the final effluent of the Kraft mill, promotes the idea of coupling more advanced membrane-type technologies (i.e., reverse osmosis or electrodialysis) and/or chemical technology (i.e., advanced oxidation processes) which may be the key to generating perspectives of sustainability of the Kraft pulp mills that are operating in countries strongly affected by climate change.

The coupling of technology to make the Kraft pulp mill processes sustainable will cause a closing of the cycle, enhancing the value of the raw material and giving space to the integration of the operation units with each other, integrating production of individual products into a multi-product biorefinery. This productive change will generate incentives for the development of biotechnology of lignocellulosic compounds in general, promoting an integration between natural resources, economy, and the environment.

**Author Contributions:** Conceptualization, G.V. and B.P.; methodology, M.J.; software, G.G.; validation, M.J.; formal analysis, M.J.; investigation, M.J.; resources, G.V.; data curation, G.G.; writing—original draft preparation, B.P., G.G., Y.G., and G.V.; writing—review and editing, G.G. and G.V.; visualization, G.G.; supervision, G.V.; project administration, G.V.; funding acquisition, G.V. All authors have read and agreed to the published version of the manuscript.

**Funding:** This research was funded by ANID/FONDAP/15130015.

**Acknowledgments:** This work was supported by ANID/FONDAP/15130015.

**Conflicts of Interest:** The authors declare no conflict of interest.

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
