# Peer review of "Minimization of Environmental Impact of Kraft Pulp Mill Effluents: Current Practices and Future Perspectives towards Sustainability"

_sustainability, doi:10.3390/su13169288_

Round 1

Reviewer 1 Report

The work of Vidal et al. proposes to review the treatment of
Kraft pulp mill effluents and their impact regarding the current practices and future perspectives towards sustainability of mills under climate change.

The title is good.

The abstract is concise and provides sufficient information.

I am seriously disappointed that not a single mention of Candida utilis (Cyberlindnera jadinii, aka Torula) was made, an organism that is most abundant in such lignin rich environments. Please introduce such information in your paper as it is an important microbiological factor with economical implications.

"3. Evaluation of biological activity effects of the Kraft mill effluents"

The state of  Zoobenthos should also be included in the ecotoxicological assay

"Table 2"

High levels of Vitelogenin mean fish get in contact with estrogen-like compounds (such as Bisphenol A), perhaps some phenolic compounds can also bind to estrogen receptors, please look deeper into that and provide such information in your review.

A few details regarding phytoremediation such as Lemna minor [1] uses and phytomicrobiome [2] are necessary.

References

1. Nguyen Thi Kim Oanh, Bengt-Erik Bengtsson, Toxicity to Microtox, micro-algae and duckweed of effluents from the Bai Bang paper company (BAPACO), a Vietnamese bleached kraft pulp and paper mill, Environmental Pollution, Volume 90, Issue 3, 1995, Pages 391-399, ISSN 0269-7491, https://doi.org/10.1016/0269-7491(95)00008-F.

2. FENDRIHAN S, POP CE. Biotechnological potential of
plant associated microorganisms. Rom Biotechnol Lett. 2021; 26(3): 2700-2706. DOI: 10.25083/rbl/26.3/2700-2706

Author Response

The answers are attached in the file.

Reviewer 2 Report

The article is interesting. However, several editorial changes should be made before its publication. First of all, its layout needs improvement.  It should contain Introduction, Methodology, Results, Discussion, and Conclusion. This also applies to the Review articles. In the Introduction section, the background of the issue should be presented and the purpose of the study should be stated at the end. In the Methodology section, you should state which databases were used for selecting articles and based on which keywords.  Later, the results should be presented.  In the next part, there should be located Discussion that is more general in nature.  For example, the current Chapter 6 could be included in this section.

Conclusions should be drawn only based on the presented results.

Detailed comments:

Line 30 and others. There should be no space between figures and %. Please correct throughout the text. Similarly with '.../...' e.g. line 88.

Line 61. Substitution should be written with a lowercase letter.

Did the Authors elaborate on Figures 2 and 3?

Line 173. Unnecessary spaces between citations.

Author Response

The answers are attached in the file.

Reviewer 3 Report

This is an interesting manuscript (review) that aims to provide insight into the impact of wastewater and even insufficiently treated from Kraft pulp mills. The authors explain the formation of wastewater, its chemical composition, the negative impact on the living biological community, give an overview of conventional and advanced treatment technologies and finally a review of sustainable development and reuse of treated water.

Since there are already published review papers on this topic, the authors should clearly emphasize in the manuscript what their contribution is in relation to the already published review papers. How did the authors contribute to solving this problem, apart from the one mentioned in Chapter 6. Also, I am of the opinion since this is a review paper - the authors should look at this issue at a global level. For example, where in the world is the most industry of this type, which processing methods do they use. Differences in the treatment of these types of wastewater in developed and underdeveloped countries. I think this would bring a big novelty to this article.

Specific comments:

Line 14: „Craft mill“ some specific or generaly Craft mills

Line 40: Define the meaning of an expression „log Kow“

In Figure 1 replace BOD with BOD5 as in the title of the Figure

Line 188: Full Stop in a sentence

Lines 234, 236, 267, 269: maning of abbreviations: ORL, CL50, LOEC, F/M ratio, MBBR - in these places they are first mentioned in a sentence without prior explanation of the abbreviation

Lines 293: „meanwhile for TOC is 30 - 70% for TOC,“- sentence construction

Line 326: change „Regarding“ to regarding

Author Response

The answers are attached in the file.

Reviewer 4 Report

The manuscript entitled “Minimization of the environmental impact of Kraft pulp mill effluents: Current practices and future perspectives towards sustainability” provided a detailed literature of Kraft pulp mill effluents and their impact regarding the current practices and future perspectives. This review clearly identified major achievements in the field in recent years, major research questions, or future research needs, which are key aspects. However, there are some concerns and points which must be improved. Authors must work on the following points:

-Abstract should be rewritten by detailing the aim and concept of the review with state-of-art in one sentence maybe..

-The abstract of a good journal paper always ends outlining the benefits of the literature findings and recommendations as a way forward. The manuscript is missing such 1-2 lines in the abstract.

-Provide significant words which are more relevant to the work in a logical sequence as ‘keywords’. Also, use keywords that are not present in the title.

-Introduction is very general and needs to be elaborative to explore the actual philosophy. Authors have done through literature survey and have presented the past works. But, what kind of innovation will be brought to the literature with this article? therefore, the state-of-art should be clearly specified in detail in the Introduction part. Hypothesis should be given. How this work is different from the available literature?

- What is the current level of understanding in relation to kraft pulp mill effluents and their impact? What are the knowledge gaps?. These should be included in the introduction section.

- The last paragraph or closing lines of the introduction section always highlight the novelty aspects of the study with the clear aim of the study and the importance/significance of the study findings. 

-It is also recommended to discuss and explain what should be the appropriate policies based on the findings of this study. Also, the literature should be further elaborated to show how they could be used for real applications. 

* Authors should also address challenges in the current research and recommendations, before the conclusion.

-Conclusion: pls. conclude with more focus on the major outcomes of the paper.

Author Response

The answers are attached in the file.

Round 2

Reviewer 1 Report

The authors have revised accordingly and I find this paper suitable for publication.

Congrats for a work well done!

Author Response

Thank you very much

Reviewer 2 Report

The attached file with the authors' response does not refer to my review. In the revised manuscript authors did not change so much. For that reason, the presented article needs to be improved.

Author Response

Answer in attached file

Reviewer 3 Report

The authors accepted and corrected the minor comments, however the main comments I made were not introduced in the paper although the authors stated: Answer: „Thank you very much for the comments. We have emphasized with the resolution of problems of the Kraft pulp mill through more specific technologies explained in greater depth in Table 4.“

Therefore, I ask the authors to carefully read the comments one more time and try to improve the quality of the manuscript.

I repeat the comments as follows:

  1. Since there are already published review papers on this topic, the authors should clearly emphasize in the manuscript what their contribution is in relation to the already published review papers.
  2. How did the authors contribute to solving this problem, apart from the one mentioned in Chapter 6.
  3. Also, I am of the opinion since this is a review paper - the authors should look at this issue at a global level. For example, where in the world is the most industry of this type, which processing methods do they use. Differences in the treatment of these types of wastewater in developed and underdeveloped countries. I think this would bring a big novelty to this article.

Make changes in the paper.

The comment was:

Line 188: Full Stop in a sentence

I wrote the line wrong, I meant line 88, in the new version it is line 89

The conclusion is incorrectly numbered, number 7

Author Response

Answer in attached file

Reviewer 4 Report

The author presented a revision to previous comments in the revised manuscript. There are still some points which must be improved. Authors should work on the following points:

-The manuscript needs language editing since it is difficult to understand what the authors are trying to convey in places.

- The introduction section is required to be improved. The introduction of the paper must be extended and reformulated in order to provide a more comprehensive approach.

Author Response

Answer in attached file

Round 3

Reviewer 2 Report

The article can be published in present form.

Reviewer 3 Report

The authors have made changes to the manuscript, therefore I accept the manuscript in this form.